Survival prediction in gliomas based on MRI radiomics combined with clinical factors and molecular biomarkers

Hao Min 1 2
Yan Junyu 1 3 4
Wang Xiaochun 1
Tan Yan 1
Zhang Hui 1 zhang_hui@sxmu.edu.cn
Yang Guoqiang 1 yangguoqiang@sxmu.edu.cn
1 Department of Radiology, The First Hospital of Shanxi Medical University , Taiyuan, Shanxi , China
2 College of Medical Imaging, Shanxi Medical University , Taiyuan, Shanxi , China
3 Academy of Medical Sciences, Shanxi Medical University , Taiyuan, Shanxi , China
4 Department of Health Statistics, School of Public Health, Shanxi Medical University , Taiyuan, Shanxi , China
Shibuya Kenichi
Electronic publication date: 2025 Aug 20
Publication date: 2025
Volume: 13
Electronic Location ID: e19906
Received 2024 Nov 15; Accepted 2025 Jul 23
Copyright: © 2025 Hao et al.
Copyright year: 2025
Copyright holder: Hao et al.
License: This is an open access article distributed under the terms of the Creative Commons Attribution License, which permits unrestricted use, distribution, reproduction and adaptation in any medium and for any purpose provided that it is properly attributed. For attribution, the original author(s), title, publication source (PeerJ) and either DOI or URL of the article must be cited.
License URL: https://creativecommons.org/licenses/by/4.0/

Keywords: Glioma, Survival prediction, MRI, Radiomics, Nomogram

Funding: National Natural Science Foundation of China U21A20386, 82071893, 82371941 Applied Basic Research Project of Shanxi Province 202303021211204 This work was supported by the National Natural Science Foundation of China (U21A20386, 82071893, 82371941), the Applied Basic Research Project of Shanxi Province (202303021211204). The funders had no role in study design, data collection and analysis, decision to publish, or preparation of the manuscript.

==============================
Background

To investigate the practicability of a radiomics signature combined with clinical factors and molecular biomarkers for predicting overall survival (OS) in glioma patients.

Methods

Training (n = 331) and internal validation (n = 83) sets were retrospectively collected from the Cancer Image Archive/The Cancer Genome Atlas (TCIA/TCGA), and 165 patients from our hospital for an external validation set. The least absolute shrinkage and selection operator (LASSO) was developed to select features. A radiomics model was established for predicting OS based on contrast-enhanced T1-weighted imaging (CE-T1WI) and T2 fluid attenuated inversion recovery (T2FLAIR) images. The risk stratification value of the radiomics signature was explored using Kaplan-Meier survival analysis and the log-rank test. The integrated prediction model with selected clinical factors, molecular biomarkers, and radiomics features was constructed through multivariate Cox regression analysis. Radiomics prognostic performance and benefit were assessed for all cohorts.

Results

The radiomics signature based on the combined sequences indicated exceptional predictive ability for OS in three cohorts and stratified glioma patients significantly into high-risk and low-risk groups (P < 0.0001). A nomogram incorporating O6-methylguanine-DNA-methyltransferase (MGMT), isocitrate dehydrogenase (IDH), pathological grade, age, and radiomics signature showed excellent evaluation performance and good calibration for predicting OS in the training (C-index = 0.774), internal (C-index = 0.750), and external (C-index = 0.776) validation cohorts.

Conclusion

The radiomics signature demonstrates superior predictive performance for OS in glioma patients and significant subgroup risk stratification efficiency. Moreover, the comprehensive model combining clinical factors, molecular biomarkers, and radiomics features further achieves a robust assessment of survival prognosis.

Introduction

Gliomas are the most prevalent primary malignant tumors in the central nervous system, accounting for approximately 30% of all primary brain tumors, with a 5-year overall survival rate of no more than 35% (Schaff & Mellinghoff, 2023). Low-grade gliomas are characterized by relatively indolent growth, demonstrating a median overall survival (OS) ranging between 4 and 13 years (Ostrom et al., 2015). In contrast, grade 3 gliomas demonstrate a significantly shorter median OS of 3 years (Bleeker, Molenaar & Leenstra, 2012). Despite receiving the same comprehensive therapeutic regimen, which includes surgical resection, radiotherapy, and adjuvant chemotherapy with temozolomide (TMZ), the long-term prognosis of patients remains unfavorable (Jakacki et al., 2016; Malmstrom et al., 2012). Especially for glioblastoma after standard postoperative treatment, the median survival is only 12–15 months (Ostrom et al., 2017). Patient age, tumor topography, molecular profile, and the extent of surgical resection are correlated with disease trajectory and survival status (Molinaro et al., 2019; Śledzińska et al., 2021). Even among patients with the same histopathological type, survival outcomes could exhibit marked divergence attributable to tumorigenic molecular mechanisms, including distinct somatic gene mutations, epigenetic remodeling processes, and pathological microvascular proliferation patterns (Berzero et al., 2021; Ghosh et al., 2025; Hu et al., 2017). Consequently, the investigation of biomarkers associated with prognostic survival in glioma may critically facilitate the formulation of molecularly targeted approaches and personalized clinical management strategies.

The 2021 WHO classification of central nervous system tumors includes molecular parameters other than histological features to provide effective information for the evaluation of tumor prognosis (Louis et al., 2021; Śledzińska et al., 2021). Currently, isocitrate dehydrogenase (IDH) and O6-methylguanine-DNA methyltransferase (MGMT) are the most common molecular biomarkers associated with glioma progression and prognosis (Molenaar et al., 2014; Valiyaveettil et al., 2018). Pathological detection of these genotypes requires sufficient tumor specimens obtained through puncture or surgical removal and expensive instruments (Beiko et al., 2014; Hegi et al., 2005). As the preferred inspection method for intracranial tumors, magnetic resonance imaging (MRI) can provide microstructure and heterogeneity information on gliomas non-invasively (Henson, Gaviani & Gonzalez, 2005).

Radiomics refers to converting high-dimensional image features into mineable data, allowing for multivariate quantification of tumor traits and the microenvironment (Gillies, Kinahan & Hricak, 2016). Previous radiomics studies have revealed that radiomics features may be considered individual imaging biomarkers for predictive and prognostic details of glioma (Choi et al., 2020; Kickingereder et al., 2016; Lu et al., 2020). Or rather, radiomics has been used to distinguish molecular subtypes (Wang et al., 2022; Zhang et al., 2023), diagnose pathological grade (Du et al., 2023; Wang et al., 2023), appraise response to treatment (Li et al., 2023), predict OS of patients (Li et al., 2022; Tixier et al., 2019), and improve the risk stratification performance (Kickingereder et al., 2018). In the field of survival prediction for gliomas, current methodologies predominantly employ machine learning and deep learning techniques to extract quantitative radiomic features from CE-T1WI and T2FLAIR sequences, integrating these imaging biomarkers with clinical parameters and molecular markers to develop composite prognostic models (Lu et al., 2020; Wan et al., 2025; Yan et al., 2021). However, these studies are primarily based on single-center datasets, with model generalizability remaining unverified across independent external institutions with varying imaging protocols and data sources. In this study, we constructed radiomics models leveraging publicly available datasets and performed rigorous external validation using our institutional imaging cohort, thereby evaluating the generalizability of the models across heterogeneous data sources. Furthermore, stratified analyses were conducted across clinical (gender, grade, and age) and molecular subgroups (IDH status and MGMT promoter methylation status), comprehensively elucidating the prognostic risk stratification capacity of the radiomic signature.

Hence, this study aimed to construct a radiomics signature based on CE-T1WI and T2FLAIR sequences and further develop a nomogram incorporating the radiomics signature, clinical features, and molecular predictors for evaluating 1-, 2-, and 3-year OS. The generalization of this combined model was validated in an independent external validation cohort.

Materials and Methods

Patients

The patient data used for this study was approved by the Institutional Review Board of First Hospital of Shanxi Medical University, with the approval number: 2021 K-K073. For another TCGA/TCIA database, since all information on the patients was accessible for public download, no institutional review board permission was required.

Public datasets can be downloaded from The Cancer Imaging Archive (TCIA) official website (https://www.cancerimagingarchive.net/collections/). A total of 709 participants from the TCGA/TCIA project were enrolled, including TCGA-LGG, TCGA-GBM, and UCSF-PDGM datasets as the training cohort and internal validation cohort. Additionally, 326 patients with pathologically diagnosed grade 2–4 glioma were retrospectively collected between October 2011 and April 2019 from the First Hospital of Shanxi Medical University (FHSXMU) and Shanxi Provincial People’s Hospital (SPPH) as an external validation cohort. All patients determined in this study met the following criteria: (a) pathological confirmation of Grade 2–4 glioma post-surgery; (b) availability of complete preoperative MRI data, including CE-T1WI and T2FLAIR images; (c) determination of tumor molecular markers IDH and MGMT; (d) a follow-up time exceeded 2 years or endpoint events occurred. The data for the 414 patients from TCGA/TCIA and the 165 patients from our hospital included in the study are shown in Table S1. Overall survival was calculated from the moment of postoperative pathological diagnosis to death or the last follow-up if patients were still alive. The detailed patient inclusion process is shown in Fig. 1.

Figure 1 Flowchart of the glioma patient enrollment.

MRI protocol

Preoperative MRI images were acquired on a 3.0T scanner (Signa HDxt, GE Healthcare, Waukesha, WI, USA) equipped with an eight-channel matrix coil. The sequence parameters included CE-T1WI (TR/TE, 195/4.76 ms; FOV, 240 × 240 mm2; slice thickness/spacing, 5.0/1.5 mm; matrix, 256 × 256) and T2FLAIR (TR/TE, 8,000/95 ms; TI, 2,000 ms; FOV, 240 × 240 mm2; slice thickness/spacing, 5.0/1.5 mm; matrix, 256 × 256) images. The CE-T1WI was carried out after the injection of 0.1 mmol/kg gadolinium contrast agent.

IDH genotyping and MGMT promoter methylation detection

For patients in the TCGA/TCIA queue, IDH mutation status and MGMT methylation data were downloaded from TCGA and cBioPortal (http://www.cbioportal.org/). Institutional specimens underwent deparaffinization followed by microdissection to ensure ≥80% tumor cellularity. DNA was extracted using the Simlex OUP® FFPE DNA kit. IDH1/2 mutations were analyzed by Sanger sequencing after PCR amplification (ABI9700) using specific primers: IDH1-F: 5′-CGGTCTTCAGAGAAGCCATT-3′ and R: 5′-GCAAAATCACATTATTGCCAA-3′, and for IDH2-F: 5′-CAAGAGGATGGCTAGGCGAG-3′ and R: 5′-CAAGCTGAAGAAGATGTGGAAAAG-3′.

Pyrosequencing was used to assess MGMT methylationb (Quillien et al., 2014). The extracted DNA was modified with bisulfite using the BisulFlashTM DNA modification kit (Epigentek, Farmingdale, NY, USA). PCR amplification was performed using the DRR006 kit (Takara, Shiga, Japan) with a reaction volume of 40 μl. PCR amplification (DRR006 kit, 40 μl) conditions: 94 °C/2 min; 50 cycles of 94 °C/20 s, 55 °C/20 s, 72 °C/20 s; final extension 72 °C/5 min. Pyrosequencing of the PCR products was performed on the PyroMark Q96 system (Qiagen, Hilden, Germany) according to the provided instructions. Ten CpG sites were quantitatively analyzed within the MGMT promoter, with methylation thresholds defined as ≥8% (methylated) and <8% (unmethylated) based on averaged values.

Image preprocessing and tumor ROI segmentation

UCSF-PDGM MRI data were preprocessed using an open-source deep-learning tool to remove skull structures. Images were aligned and resampled to 1 mm isotropic resolution in T2FLAIR space through nonlinear registration. Tumor regions (enhancing tumor, necrosis, and edema) were automatically segmented using Brain Tumor Segmentation (BraTS) Challenge-based algorithms. Detail the segmentation methodologies applied to the UCSF-PDGM dataset in the PDF referenced by the DOI: https://doi.org/10.48550/arXiv.1811.02629. The code for brain tumor segmentation algorithms based on the BraTS Challenge can be found in the GitHub project brats-unet: UNet for brain tumor segmentation (https://github.com/icerain-alt/brats-unet). We selected the contrast-enhanced tumor regions for feature extraction and subsequent analysis. The preprocessing pipeline for the CE-T1WI and T2FLAIR images from TCGA-LGG, TCGA-GBM, and our institutional data involved spatial resolution normalization through resampling to isotropic 1 mm3 voxel dimensions. We performed z-score normalization on the image grayscale values, resulting in a mean of 0 and a standard deviation of 1 (Carré et al., 2020). The T2FLAIR images were co-registered with the corresponding CE-T1WI images by affine transformation using the Oxford Centre for Functional MRI of the Brain (FMRIB) Linear Image Registration Tool (FLIRT) from the FMRIB Software Library (FSL). All CE-T1WI and T2FLAIR MRI images underwent resampling and normalization using Matlab. Tumors were manually segmented on the axial thin-section 2D slices by two radiologists with 10 and 15 years of experience using ITK-SNAP software (http://www.itksnap.org). For enhanced tumors, the region of interest (ROI) was described based on the enhanced edge on CE-T1WI images. For unenhanced tumors, the tumor signal on CE-T1WI images was lower compared to the peritumoral edema area, and the ROI was determined based on the scope of unenhanced lesions on CE-T1WI images. The final ROI was verified by a senior radiologist with 20 years of experience. The intraclass and interclass correlation coefficients (ICC) were utilized to assess the stability and repeatability of radiomic features. To evaluate intergroup consistency, the first two radiologists independently depicted 30 arbitrary gliomas. Two weeks later, the first physician re-delineated the same 30 cases to estimate intragroup consistency. The flowchart of this study is illustrated in Fig. 2.

Figure 2 Workflow diagram of the radiomics pipeline.

(I) Image processsing and ROI segmentation: register T2FLAIR to CE-T1WI images, resample all images, and perform intensity normalization. Following preprocessing, manually delineate ROI slice-by-slice on the CE-T1WI images. (II) Feature extraction: features were extracted using the open-source software FAE based on PyRadiomics. Feature types include shape, first-order and texture, and nine filter transformations. (III) Feature selection: first, retain features with ICC > 0.75. Then, perform univariable Cox regression analysis to select features significantly associated with OS. Finally, apply the LASSO algorithm for dimensionality reduction to obtain the optimal features. (IV) Model building and validation: Integrate radiomic signatures with clinical molecular models to develop a combined model, visualize the model using a nomogram, evaluate the consistency between the predicted probability of OS and actual observations through calibration curves, and assess the clinical net benefit of the nomogram using decision curve analysis.

Radiomics feature extraction and selection

A total of 1,781 radiomics features were independently extracted from the tumor regions of CE-T1WI and T2FLAIR MRI images using the open-source software FAE (https://github.com/salan668/FAE) based on PyRadiomics. These features included 14 shape features, 18 first-order features, 75 texture features (consisting of 24 GLCM features, 14 GLDM features, 16 GLRLM features, 16 GLSZM features, and five NGTDM features) of the original images, 144 first-order features and 600 texture features of wavelet transform, 18 first-order features and 75 texture features drawn on square, square root, gradient, laplace, exponent, logarithm, 2D transformation, and 279 first-order and texture features based on 3D transformation. Features with an ICC higher than 0.75 were used for subsequent analysis. Z-score was used to normalize the CE-T1WI and T2FLAIR feature matrices before feature screening in the training cohort, internal and external validation cohorts. Initially, univariate Cox proportional hazards regression was conducted to filter out variables demonstrating no significant correlation with OS in the training cohort (P > 0.05). Subsequently, the least absolute shrinkage and selection operator (LASSO) method was performed to select significant features that were strongly associated with patient overall survival.

Construction of clinical-molecular prediction model

Univariate Cox regression analysis was performed to analyze the correlation between clinical-molecular characteristics and OS, and those with P < 0.05 were included in the multivariate Cox regression analysis. Furthermore, independent risk predictors of OS were selected for the construction and validation of clinical-molecular models.

Radiomics signature construction and validation

These specific features were integrated into the radiomics signature, and the radiomics risk score was calculated based on the linear combination of weight coefficients assigned to each selected feature. Three sets of radiomics models (CE-T1WI, T2FLAIR, and combined-sequences) were established. Firstly, the correlation between the radiomics riskscore and OS was profiled by univariate Cox regression. The C-index of radiomics models was calculated, and the area under the curve was used to evaluate the diagnostic efficacy of the radiomics signatures in predicting 1-, 2-, and 3-year survival. The optimal radiomics signature was attained. The patients were divided into high-risk and low-risk groups according to a fixed cutoff value for Kaplan-Meier survival analysis. Finally, to demonstrate the incremental value of radiomics features, risk stratification was performed with the radiomics signature in various clinical and molecular subgroups. The detailed computational workflow is as follows:

Riskscore=(coeffeature1×feature1)+(coeffeature2×feature2)+⋯(coeffeaturen×featuren).

The coefficients were obtained from the multivariate Cox regression model of selected features.

Construction and evaluation of nomogram

A nomogram integrating the radiomics signature and clinical molecular risk factors independently predicted OS in glioma patients. The net reclassification index (NRI) was used to evaluate the enhancement of the radiomics signature to the clinical-molecular model. The calibration curve was used to assess the consistency between the predicted probability of survival and the actual survival. Decision curve analysis (DCA) compared the net benefits of the radiomics model, clinical-molecular model, and combined model at different threshold probabilities, which confirmed the clinical utility of the nomogram.

Statistical analysis

All statistical analyses were performed using R software, version 4.2.3 (http://www.R-project.org). The R packages used in this analysis were as follows: the “glmnet” package was used for LASSO Cox regression; Kaplan-Meier survival analysis and multivariate Cox regression were done using the “survival” package and the “survminer” package. The “timeROC” package was used to plot ROC curves; Nomogram and calibration curve were performed using the “rms” package; C-index calculation was implemented using the “Hmisc” package. A two-sided p-value of 0.05 indicated statistical significance.

Results

Patient clinical characteristics

The clinical characteristics of all patients are shown in Table 1. For the TCGA/TCIA dataset, 414 patients were split into a training set (n = 331) and an internal validation set (n = 83) at an 8:2 ratio. The median OS of the training set and the internal validation set was 498 days (95% CI [443.49–552.51]) and 561 days (95% CI [402.29–719.71]), respectively. In the external validation set (n = 165), 103 patients died and 62 patients survived by the end of follow-up. The median follow-up duration was 1,245 days (95% CI [1,118.21–1,371.79]), with a median OS of 777 days (95% CI [628.85–925.15]). There were significant differences in the distribution of age, gender, pathological grade, IDH, and OS among the three groups of cohorts (P < 0.05), while only MGMT showed no statistically significant difference (P > 0.05).

Table 1 Clinical characteristics of the three cohorts.

Characteristic	Training
cohort (n = 331)	Internal validation
cohort (n = 83)	External validation cohort (n = 165)	P value	
Age (year)	58 (49–67)	59 (47–70)	52 (40.50–61.00)	<0.001	
Gender				0.005	
Male	176 (53.17)	56 (67.47)	75 (45.45)		
Female	155 (46.83)	27 (32.53)	90 (54.55)		
Grade				<0.001	
2	25 (7.55)	4 (4.82)	52 (31.52)		
3	36 (10.88)	10 (12.05)	57 (34.55)		
4	270 (81.57)	69 (83.13)	56 (33.94)		
IDH status				<0.001	
Wild type	267 (80.66)	67 (80.72)	86 (52.12)		
Mutantion	64 (19.34)	16 (19.28)	79 (47.88)		
MGMT promoter				0.235	
Unmethylation	99 (29.91)	24 (28.92)	61 (36.97)		
Methylation	232 (70.09)	59 (71.08)	104 (63.03)		
Overall survival
(day)	498 (248–958)	561 (213–930)	777 (329–1,127.50)	0.008	

Construction and validation of a clinical-molecular prediction model

According to univariate Cox regression analysis, age, pathological grade, IDH, and MGMT were significantly correlated with OS (P < 0.05), while gender was not statistically significant (P > 0.05). Multivariate Cox regression analysis further demonstrated that age, pathological grade, IDH, and MGMT remained as independent risk factors for glioma prognosis (P < 0.05). The specific P-values and HR are shown in Table 2. The C-index of the predictive model constructed based on clinical and molecular factors in the training set, the internal validation set, and the external validation set was 0.719, 0.713, and 0.689, respectively.

Table 2 Cox regression analysis of all clinical and molecular factors in the training cohort.

Variable	Univariate analysis	Multivariate analysis	
HR (95% CI)	P	HR (95% CI)	P	
Gender					
Male	1.000				
Female	0.996 [0.780–1.270]	0.972			
Age	1.052 [1.041–1.062]	<0.001	1.036 [1.025–1.048]	<0.001	
Grade					
2	1.000		1.000		
3	5.104 [1.875–13.89]	0.001	4.476 [1.556–12.876]	0.005	
4	13.381 [5.246–34.13]	<0.001	3.864 [1.316–11.341]	0.014	
IDH					
Wild type	1.000				
Mutation	0.114 [0.068–0.189]	<0.001	0.276 [0.144–0.531]	<0.001	
MGMT					
Unmethylation	1.000				
Methylation	0.526 [0.406–0.681]	<0.001	0.702 [0.541–0.911]	0.008	

Radiomics signature construction and validation

As a whole, 19, 16, and 31 features were screened using LASSO Cox regression from CE-T1WI, T2FLAIR, and combined sequence images, respectively. The process of combined-sequences filtering features is shown in Fig. 3. In the training set and internal validation set, the combined sequences radiomics model demonstrated a higher area under the curve (AUC) (0.84 & 0.84, 0.84 & 0.91, 0.86 & 0.91) in predicting 1-, 2-, and 3-year survival compared to the CE-T1WI radiomics model (0.84 & 0.77, 0.85 & 0.80, 0.84 & 0.82) and the T2FLAIR radiomics model (0.77 & 0.65, 0.77 & 0.70, 0.76 & 0.73). Additionally, the multiparametric radiomics signature showed a better C-index compared to the single sequence model, making it the optimal radiomics model (Table 3). Hazard ratio (HR) values with 95% confidence interval (CI) and P values of each selected feature of the combined sequences radiomics model are shown in Fig. S1. The Spearman correlation heatmap illustrates the relationships between the selected significant radiomics features and clinical parameters in the training cohort (Fig. 4). Among the five clinical parameters, gender and MGMT status demonstrated stronger positive correlations with most radiomics features, while age, tumor grade, and IDH status exhibited weaker associations. The process for constructing the multiparametric radiomics signature was summarized in the supplementary documentation. The patients were split into high-risk and low-risk groups based on the median risk score (1.143) of the training set in three cohorts (Fig. 5).

Figure 3 Feature selection based on LASSO Cox regression.

Table 3 Performance of three radiomics models for predicting OS.

Model	Performance	C-Index (95% CI)	1-year AUC	2-year AUC	3-year AUC	
CE-T1WI	Training cohort	0.763 [0.742–0.784]	0.84	0.85	0.84	
Internal validation cohort	0.717 [0.682–0.752]	0.77	0.80	0.82	
External validation cohort	0.665 [0.648–0.682]	0.71	0.74	0.80	
T2FLAIR	Training cohort	0.715 [0.677–0.753]	0.77	0.77	0.76	
Internal validation cohort	0.656 [0.631–0.681]	0.65	0.70	0.73	
External validation cohort	0.645 [0.627–0.663]	0.68	0.74	0.78	
Combined sequences	Training cohort	0.771 [0.748–0.794]	0.84	0.84	0.86	
Internal validation cohort	0.778 [0.748–0.808]	0.84	0.91	0.91	
External validation cohort	0.711 [0.703–0.719]	0.79	0.80	0.83	

Figure 4 Spearman correlation heatmap of significant radiomics features and clinical parameters in the training cohort, darker color and larger bubble size indicate stronger correlations between variables.

Figure 5 Kaplan Meier survival curves of radiomics signature in the training cohort (A), internal validation cohort (B) and external validation cohort (C).

Sub-stratified survival analysis of the radiomics signature

To prove the valuable predictive effectiveness of the radiomics model, subgroup survival analysis was performed to evaluate differences in OS between different clinical and molecular subgroups (Figs. 6 and 7). In the training set of clinical (age, gender) and molecular (MGMT) subgroups, there were statistically significant differences between high-risk and low-risk groups. The P-values of the log-rank test were all under 0.0001. In the internal validation set, significant differences were exhibited in each subgroup, with P-values less than 0.05. In the age, gender, IDH, and MGMT subgroups of the external validation set, the radiomics signature significantly stratified high-risk and low-risk groups according to the risk score cut-off value. While in the IDH mutation group of the training set and GBM group of the external validation set, risk stratification was not achieved.

Figure 6 Stratified Kaplan–Meier analysis based on different age subgroups (younger or older), gender (male or female), and grade (GBM or Non-GBM) in the training set (A, B), internal validation set (C, D), and external validation set (E, F).

Figure 7 Stratified Kaplan–Meier analysis based on different IDH subgroup (IDH wild or mutant), MGMT subgroup (meth or unmeth) in the training set (A, B), internal validation set (C, D), and external validation set (E, F).

Nomogram establishment and evaluation

The radiomics signature and selected risk factors generated a nomogram for OS prediction (Fig. 8a). Compared to the clinical-molecular model (C-index = 0.719 and 0.689) and radiomics signature (0.771 and 0.711), the nomogram showed superior ability to predict prognosis in the training (0.774) cohort and external validation cohort (0.776). The calibration curves showed delightful agreement between the predicted values for 1-, 2-, and 3-year survival probabilities and the actual values for the combined model in the training cohort, internal validation, and external validation cohort (Figs. 8B–8D). Upon further analysis, after integrating the radiomics signature into the clinical-molecular model, the integrated model produced an NRI of 0.545 (95% CI [0.115–1.058]), improving the proportion of correct classification. The comprehensive model provided more net benefit across most threshold ranges in contrast to other models in the decision curves analysis (Fig. 8E).

Figure 8 Establishment and evaluation of the integrated model.

Nomogram integrating radiomics signature, age, pathological grade, IDH and MGMT to predict 1-, 2-, and 3-year OS for glioma (A). The calibration curves of the nomogram in the training cohort (B), internal validation cohort (C) and external validation cohort (D). Decision curves for three models based on the different diagnostic factors, showing that the radiomics-clinical-molecular model was significantly superior to the clinical-molecular model and radiomics model (E).

Discussion

In this study, the multiparameter radiomics model indicated the best diagnostic performance compared to single-sequence radiomics signatures for predicting 1-, 2-, and 3-year survival in the training cohort, internal validation cohort, and external validation cohort. Subsequently, risk stratification was carried out based on the combined radiomics model in different clinical and genetic subgroups, verifying the incremental value of radiomics features in glioma prognosis assessment. Eventually, incorporating the clinical-molecular model into the radiomics model significantly enhances the accuracy of survival probability predictions. Importantly, this comprehensive model was further validated in the external validation cohort, yielding a C-index of 0.776.

Predictors of the clinical-molecular model were formed by multivariate Cox regression analysis, including pathological grade, age, IDH, and MGMT. In contrast to previous research (Liu et al., 2018; Pease et al., 2022), our results confirmed that IDH and MGMT were simultaneously incorporated into the model, rather than only one factor. Currently, common molecular markers associated with the prognosis of glioma include IDH, MGMT, telomerase reverse transcriptase (TERT), the short arm of chromosome 1 and the long arm of chromosome 19 (1p/19q) (Ludwig & Kornblum, 2017). IDH mutation promotes tumorigenesis by affecting cell metabolism and its microenvironment, and IDH-mutant gliomas are associated with lower grade and favorable survival (Berzero et al., 2021; Hartmann et al., 2010). Methylation of the MGMT promoter is correlated with the sensitivity of glioma patients to alkyl chemotherapies (Wick et al., 2014). Sareen et al. (2022) conducted a meta-analysis on the predictive performance of biomarkers and found that five studies involving 541 GBM patients with IDH1 mutation showed a significant improvement in OS for patients with IDH1 mutation. Chai et al. (2021) performed a Kaplan-Meier survival analysis on 50 patients treated with TMZ, and the results revealed that patients with a methylation level of ≥30% had the longest progression-free survival (PFS) and OS. These findings indicate that the increased application of molecular biomarkers provides an important basis for the adjuvant diagnosis and targeted therapy of glioma.

Radiomics analysis is a valuable prediction tool for high-dimensional extraction of image features that contribute essential information in cancer diagnosis, prognosis, and treatment selection (Yip & Aerts, 2016). Multiple studies have confirmed the clinical practical value of radiomics features to predict glioma OS (Bae et al., 2018; Lao et al., 2017; Yan et al., 2021). However, the majority of radiomics studies have primarily focused on predicting the survival of GBM and lacked external validation from independent cohorts. This may impact the robustness of the models and clinical applicability. In the current study, we utilized radiomic features based on CE-T1WI and T2FLAIR sequences of grade II–IV gliomas for survival analysis. The thirty-one radiomic features consisted of 20 CE-T1WI sequences and 11 T2FLAIR sequences achieved better prediction performance than the single sequence model, which was consistent with previous finding (Tan et al., 2019). CE-T1WI holds information about the enhancement characteristics of glioma and the perfusion of peripheral blood vessels. T2FLAIR has superiority in identifying the tumor boundary and displaying the edema area. The combination of radiomics features from the two sequences further improved the survival prediction performance of glioma. Multimodal MRI technology provides comprehensive and detailed biological approach to the diagnosis and prognosis of glioma from different perspectives. Yan et al. (2021) developed a fusion model based on CE-T1WI and the apparent diffusion coefficient (ADC) to predict molecular typing and evaluate the survival prediction performance of glioma patients. The results showed that the fusion model achieved significant stratification in the molecular subgroup and optimal identification efficiency for predicting IDH. Zhang et al. (2020) found that dynamic contrast-enhanced (DCE)-MRI was superior to dynamic susceptibility contrast (DSC) imaging in identifying glioma molecular typing. In addition, the 10th percentile AUC of Ve from DCE provides the robust OS prediction effect as the best threshold.

To further verify the value of the radiomics model in predicting OS, we performed risk stratification and log rank tests with the combined-sequences radiomics model in different subgroups of clinical and molecular independent risk factors. In the five subgroups of the internal validation set, the radiomics signature significantly stratified the OS of the high-risk group and the low-risk group. Notably, the inability of radiomics models to achieve effective stratification of GBM subgroups may be attributed to two principal factors. First, the reduced sample size within GBM subgroup after stratification (accounting for only 33.9% of glioma cases) could compromised statistical power, resulting in inconsistent stratification outcomes. Second, further risk stratification in GBM remains challenging owing to its pronounced intratumoral and intertumoral heterogeneity, coupled with the characteristically poor survival outcomes. In the future, the prospective integration of expanding cohorts with more clinical and molecular profiling in the model is expected to enhance the precision of risk stratification. In a nutshell, the survival analysis results demonstrate the incremental predictive value of the radiomics model as an independent prognostic biomarker.

The nomogram visually displays the relationships between multiple predictors and visualizes the predicted probabilities of outcome events, which is widely applied in cancer prognosis (Chen et al., 2018; Gittleman et al., 2017; Liang et al., 2015). In several parallel studies (Bae et al., 2018; Liu et al., 2018), some scholars delineated a nomogram that integrates radiomics features with clinical information and has the advantage of being more accurate in predicting survival than single-modality prediction models. We carried out equal work that integrated radiomics signature and clinical-molecular risk factors (age, grade, IDH, MGMT) to build a comprehensive model. Both in the training and the validation sets, our combined model achieved a more stabilized assessment of patient outcomes. Afterward, a visual nomogram was drawn to predict the survival probability of 1-, 2-, and 3-year, and to reach the quantitative prediction of OS in glioma patients with the multivariate combination. Finally, the calibration curve and decision curve manifested multimodality integrated model have powerful clinical practicability in most situations. Our results mean that a nomogram could assist clinicians in selecting the appropriate treatment specifically for patients and empower precision treatment.

Even though the impressive results, this study still has certain limitations. First, our dataset was from a multicenter institution, and the heterogeneity of MRI imaging parameters could not be controlled. Secondly, due to the incomplete clinical data of most patients, clinical characteristics such as KPS score, surgical resection degree, and treatment plan were not included in this study. Thirdly, radiomics analysis was performed only on conventional magnetic resonance sequences, and more versatile sequences such as dynamic sensitivity contrast-enhanced (DSC) perfusion-weighted imaging functional sequences, susceptibility-weighted imaging (SWI), and diffusion tensor imaging (DTI) were not involved. Hence, with the expansion of the sample size, it is necessary to conduct multimodality radiomics analysis to capture prognostic information to improve the robustness of the model.

Conclusions

In conclusion, the radiomics signature can accurately predict the overall survival of glioma and stratify patients. The comprehensive model integrating radiomics features, clinical factors, and molecular biomarkers further enhances the prognostic prediction performance and provides a new decision-making direction for personalized diagnosis of glioma.

Supplemental Information

Supplemental Information 1 Code.

Supplemental Information 2 Translation Codebook.

Supplemental Information 3 Raw data from FHSXMU and SPPH.

Supplemental Information 4 Raw data from the TCGA-LGG, TCGA-GBM, and UCSF-PDGM datasets.

Supplemental Information 5 Clinical and characteristic table of all 579 patients.

Supplemental Information 6 Categorical data encoding.

Supplemental Information 7 The constructed radiomics signature and HR values with 95% CI and p values of each selected feature of the radiomics model.

Additional Information and Declarations

Competing Interests

The authors declare that they have no competing interests.

Author Contributions

Min Hao conceived and designed the experiments, performed the experiments, analyzed the data, prepared figures and/or tables, and approved the final draft.

Junyu Yan conceived and designed the experiments, performed the experiments, analyzed the data, prepared figures and/or tables, and approved the final draft.

Xiaochun Wang conceived and designed the experiments, performed the experiments, analyzed the data, prepared figures and/or tables, and approved the final draft.

Yan Tan conceived and designed the experiments, performed the experiments, prepared figures and/or tables, and approved the final draft.

Hui Zhang conceived and designed the experiments, authored or reviewed drafts of the article, and approved the final draft.

Guoqiang Yang performed the experiments, authored or reviewed drafts of the article, and approved the final draft.

Human Ethics

The following information was supplied relating to ethical approvals (i.e., approving body and any reference numbers):

This study at our institutional was approved by the Ethics Committee of First Hospital of Shanxi Medical University, with the approval number: 2021 K-K073.

Ethics

The following information was supplied relating to ethical approvals (i.e., approving body and any reference numbers):

The Ethics Committee of First Hospital of Shanxi Medical University approved the study (2021 K-K073).

Data Availability

The following information was supplied regarding data availability:

Data and code are available at Zenodo:

Hao, M. (2024). Original features and code of clinical prediction model [Data set]. Zenodo. https://doi.org/10.5281/zenodo.14159798.

Images are available at Zenodo:

Hao, M. (2025). UCSF-PDGM-1. Zenodo. https://doi.org/10.5281/zenodo.15299918.

Hao, M. (2025). UCSF-PDGM-2. Zenodo. https://doi.org/10.5281/zenodo.15311197.

Hao, M. (2025). UCSF-PDGM-3. Zenodo. https://doi.org/10.5281/zenodo.15322208.

Hao, M. (2025). TCGA-Image-1. Zenodo. https://doi.org/10.5281/zenodo.15323004.

Hao, M. (2025). TCGA-Image-2. Zenodo. https://doi.org/10.5281/zenodo.15324019.

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
