# Peer review of "Survival prediction in gliomas based on MRI radiomics combined with clinical factors and molecular biomarkers"

_PeerJ, doi:10.7717/peerj.19906_

## Round 0.1 · original submission · Major Revisions

Please address all the reviewer comments.

Reviewer 1 ·

Basic reporting

This manuscript titled “Survival prediction in gliomas based on MRI radiomics combined with clinical factors and molecular biomarkers” tried to utilize computer artificial intelligence method to extract magnetic resonance image features of glioma, carries out radiomics analysis using both public data-set mining and a cohort data from the author’s hospital. The authors tried to calculated a radiomic signature which can predict OS of glioma through feature screening, univariate Cox regression, the least absolute shrinkage and selection operator (LASSO) multivariate analysis. The authors also create a nomogram integrating clinical, pathological variables with radiomic signature to predict the prognosis of glioma, hoping to obtain a better prognosis model.

Experimental design

There are many items not good or with defects in the overall design idea.
First, the research objective is not clear and there is no basic hypothesis. The current research status of radiomics of glioma is not fully understood, and the research gap or problems which have been solved or have been challenged have not been in-depth described in the INTRODUCTION section, and the clinical problems tried to be solved in this research cannot be found.
Second: the METHOD section is written with crude content, without sufficient details and cannot be replicated. The process of TCGA and TCIA data mining, extraction and matching are not mentioned. Finally gathered patient data is lacking.
Third, the quality requirements of MRI images are not mentioned, the image preprocessing process is lacking, and the radiomics feature extraction and analyzing are not standardized
Fourth, Case inclusion and exclusion criteria described in the text is not proper, also about the grouping criteria. A table of baseline data is lack.
Last: Raw data provided is not sufficient, only statistics (R language) and results values can be found.

Validity of the findings

no comment

Additional comments

Figures are not well labelled & described

Annotated reviews are not available for download in order to protect the identity of reviewers who chose to remain anonymous.

Reviewer 2 ·

Basic reporting

no comment

Experimental design

no comment

Validity of the findings

no comment

Additional comments

In my opinion the paper well arranged and written, statistical analysis done and explained.
Good results and discussion.
A weak point as they mentioned some clinical and molecular data was missed.
Also there was a weak point regarding radiological assessment.

Reviewer 3 ·

Basic reporting

This study effectively created a radiomics-based nomogram to predict overall survival (OS) in Glioblastoma. The nomogram utilized a radiomics signature derived from CE-T1WI and T2FLAIR sequences, alongside clinical and molecular predictors. The multiparameter radiomics model demonstrated superior diagnostic performance compared to models using single imaging sequences. Furthermore, integrating clinical and molecular data into the radiomics model significantly increased the accuracy of survival predictions.
Literature references were great, and good background was provided.

Experimental design

No comments. Well done on the design!

Validity of the findings

The combined model was validated in an independent external validation cohort which makes sense.

Additional comments

4 minor comments:
1. The authors should provide detailed figure with a radiomics workflow: ROI segmentation -> Feature extraction -> Feature selection -> Analysis

2. The authors should also give a comparison of clinical and pathological characteristics of patients between the training and validation cohorts for OS.

3. The authors should also create a heatmap of the Spearman correlation between the significant radiomics features and clinical parameters. It shows how each biomarker is positively and negatively correlated with corresponding radiomics features.

4. The authors should also comment on the heterogeneity of MRI imaging parameters from this multicenter institution study.

Reviewer 4 ·

Basic reporting

This manuscript developed a nomogram-based prognostic tool for predicting overall survival in gliomas using a combination of publicly available and local institutional data. By using a combination of clinical, molecular, and radiomics-based features derived from standard clinical MRI, the authors were able to achieve modestly successful model performance across gliomas, with less success observed within specific diagnoses (i.e. glioblastoma). This is a well-written and largely technically sound manuscript that follows the standard practices of glioma survival prediction studies.
- A demographic table breaking down the subject composition per dataset is needed.
- Figure 5 is fairly hard to read as currently presented and contains a lot of small text that is difficult to read. Consider ways to condense these results, such as including multiple group stratifications on the same Kaplan-Meier curve.

Experimental design

- The methods used to process the MRI could be described in greater detail, particularly with regards to the resampling and normalization methods, as these are known to have downstream effects on radiomic feature calculation.
- It is a little ambiguous as written if the feature normalization and feature selection steps occurred solely in the training data or in a combination of training and validation datasets. More detail here is warranted.

Validity of the findings

- The failure for the model to distinguish between high and low risk groups within GBM is a somewhat critical issue with this model. A pan-glioma model is somewhat useful before surgical diagnosis, but itself does not preclude the need for surgical resection if GBM is suspected. Furthermore, GBMs have much worse survival rates compared to lower grade tumors, so simply discriminating between tumor grade may be driving the primary survival prediction effect seen here. Therefore, it is hard to see the value of this model within specific diagnoses as a prognostic tool, where it is likely to be used clinically. More discussion on this important caveat is warranted.

---

## Round 0.2 · Minor Revisions

Please address the remaining minor comments.

Reviewer 3 ·

Basic reporting

N/A

Experimental design

N/A

Validity of the findings

N/A

Additional comments

Satisfied with the rebuttal letter and the response

Reviewer 4 ·

Basic reporting

All comments regarding basic reporting addressed appropriately and sufficiently.

Experimental design

Most comments regarding experimental design addressed sufficiently, though the specific BRATs based models used for segmentation should be cited clearly to identify the specific model used, and the specific intensity normalization procedure should be clearly stated (z-score, histogram matching, whole brain standard deviation, etc).

Validity of the findings

All comments addressed sufficiently.

Additional comments

Aside from the small methodological points raised, this manuscript has addressed my concerns thoroughly and is sufficient in my opinion for publication.

---

## Round 0.3 · accepted · Accept

I have carefully reviewed your manuscript along with the comments provided by the reviewers.

Reviewer 4 ·

Basic reporting

All edits are sufficient. I recommend the manuscript for publication. Great work!

Experimental design

-

Validity of the findings

-